# Effects of infectious disease consultation and antimicrobial stewardship program at a Japanese cancer center: An interrupted time-series analysis

Naoya Itoh[1,2]*, Nana Akazawa[1], Eri Kanawaku[1], Hiromi Murakami[1], Yuichi Ishibana[1], Daichi Kawamura[1], Takanori Kawabata[3], Keita Mori[3], Eiichi N. Kodama[4], Norio Ohmagari[2,5,6]

1 Division of Infectious Diseases, Aichi Cancer Center Hospital, Nagoya, Japan, 2 Collaborative Chairs Emerging and Reemerging Infectious Diseases, National Center for Global Health and Medicine, Graduate School of Medicine, Tohoku University, Miyagi, Japan, 3 Clinical Trial Coordination Office, Shizuoka Cancer Center, Shizuoka, Japan, 4 Division of Infectious Diseases, International Research Institute of Disaster Science, Tohoku University and Tohoku Medical Megabank Organization, Sendai, Japan, 5 AMR Clinical Reference Center, Disease Control and Prevention Center, National Center for Global Health and Medicine, Tokyo, Japan, 6 Disease Control and Prevention Center, National Center for Global Health and Medicine, Tokyo, Japan

* itohnaoya0925@ybb.ne.jp

**Data Availability Statement:** All relevant data are within the paper and its Supporting Information files.

## Abstract

In cancer patients, appropriate diagnosis and management of infection are frequently challenging owing to subtle or atypical presentation. We investigated the effectiveness of infectious disease (ID) consultations and the Antimicrobial Stewardship Program (ASP) in a Japanese cancer center. This 36-month-period, single-institution, interrupted time series analysis was retrospectively conducted during April 1, 2018–March 31, 2021, to evaluate a two-phase intervention: Phase 1 (notification of antimicrobials by the infection control team) and Phase 2 (establishing an ID consultation service and implementing ASP). Among 32,202 patients hospitalized, 22,096 and 10,106 hospitalizations occurred at baseline and during intervention period, respectively. The Antimicrobial Stewardship Team (AST) provided feedback on specific broad-spectrum antimicrobials in 913 instances (347 appropriate [38%]; 566 inappropriate [62%]), and 440 ID consultations were completed, with a 75% overall acceptance rate for AST suggestions. In Phase 2, monthly carbapenem days of therapy (CAR-DOT) decreased significantly, and narrow-spectrum antibiotic usage increased significantly in both trend and level; monthly DOT of antipseudomonal agents decreased significantly in trend. The results of these analyses of antimicrobial use are consistent with the DOT-based data based on antimicrobial use density (AUD). The total number of inpatient specimens increased significantly; the trend of multidrug-resistant *Pseudomonas aeruginosa* and methicillin-resistant *Staphylococcus aureus* infections decreased, without changes in the incidence of other resistant organisms, all-cause in-hospital mortality, and length of stay. Actual and adjusted CAR purchase costs per patient-day decreased without significant changes in the actual and adjusted purchase cost per patient-day for all intravenous antimicrobials. Combining ID consultation and ASP reduced carbapenem use without

**Funding:** The author(s) received no specific funding for this work.

**Competing interests:** The authors have declared that no competing interests exist.

negative patient outcomes. Their implementation could facilitate establishment of safe cancer treatment facilities in Japan and improve prognosis in cancer patients.

## Introduction

In recent years, antimicrobial resistance (AMR) has become a global concern and the implementation of AMR-prevention measures has become an urgent requirement for medical institutions [1]. Although carbapenem-resistant *Enterobacteriaceae* (CRE) and carbapenem-resistant *Pseudomonas aeruginosa*, which are increasing worldwide, account for <1% of drug-resistant bacteria in Japan, the resistance rates of *Escherichia coli* to third generation cephalosporins and fluoroquinolones are showing an upward trend, and the rate of methicillin-resistant *Staphylococcus aureus* (MRSA) remains high [2]. Patients with cancer, especially those with hematologic malignancies or severe neutropenia and those who undergo hematopoietic stem cell transplantation, frequently develop serious infections and receive multiple rounds of antimicrobial therapy during the progression of their underlying disease [3, 4]. In patients with cancer, appropriate diagnosis and management of infection are frequently challenging due to the subtle or atypical presentation of infections [5, 6]; therefore, substantial amounts of antimicrobials are used in these patients [7]. Carbapenems (CAR) are commonly prescribed for infections in patients with malignancies. Physicians are reluctant to discontinue or de-escalate antimicrobial therapy within the optimal treatment period for these patients [4, 8]. There is a consensus that antimicrobial use places significant antimicrobial pressure not only on the patient's normal flora, but also potentially on the surrounding environment, which significantly contributes to the development of resistant bacteria [9, 10]. For example, patients with CRE infections are at high risk for adverse outcomes from invasive infections and can spread resistant genes within healthcare facilities. Carbapenemase-producing *Enterobacteriaceae* (CPE) outbreaks have been reported in cancer centers across Japan, complicating infection control and treatment of infected patients [11]. Most studies on the Antimicrobial Stewardship Program (ASP) were conducted in the United States or European countries, and few studies have evaluated the effectiveness of ASP in other regions [12]. While there are insufficient studies globally that have addressed the role of ASPs specifically in patients with cancer [7, 8], there have clearly been more opportunities to provide guidance on appropriate antimicrobial prescription in patients with cancer than in most other patient populations and settings. However, there are no previous reports on ASP at cancer centers in Japan. Akazawa et al. [12] and Matono et al. [13] reported that an ASP intervention in a tertiary care facility in Japan reduced the use of broad-spectrum antimicrobials without negative outcomes. Thus, we conducted a retrospective study to assess the impact of the introduction of the Antimicrobial Stewardship Team (AST) and infectious disease (ID) consultation in a Japanese cancer center.

## Materials and methods

### Setting

This study was conducted at the 500-bed tertiary care Aichi Cancer Center (ACC) Hospital in Aichi, Japan. This hospital with 23 clinical departments, admits ~11,000 patients annually. The Department of Infectious Diseases is managed by an ID physician. The 15 departments responsible for inpatients are: Plastic and Reconstructive Surgery, Hematology and Cell Therapy, Thoracic Surgery, Thoracic Oncology, Gastroenterological Surgery, Gastroenterology, Orthopedic Surgery, Head and Neck Surgery, Breast Oncology, Neurosurgery, Urology,

Gynecologic Oncology, Radiation Oncology, Diagnostic and Interventional Radiology, and Clinical Oncology. The AST includes one ID physician, one pharmacist, two clinical laboratory technicians, and one infection control nurse.

## Study design

This research was a single-institution retrospective interrupted time series analysis conducted during a 36-month period from April 1, 2018, to March 31, 2021. All data for this study were obtained from the ACC Hospital database, including microorganism data from the microbiology laboratory, prescription data from the pharmacy department, patient data from the AST, and a chart review.

**ASP interventions and ID consultations.**   The intervention evaluated in this study was implemented in two phases:

*Phase 1*: Antimicrobial notification by the infection control team (ICT) (April 1, 2018, to March 31, 2020)

*Phase 2*: Establishing an ID consultation service and implementing the ASP (April 1, 2020, to March 31, 2021)

**Phase 1: Antimicrobial notification by the ICT (April 1, 2018, to March 31, 2020).** Before introduction of the ID consultation and ASP, the ICT implemented a notification policy (verification of the reason for use only, and the duration of treatment), without an intervention, for specific antimicrobial usage. The antimicrobials included CAR (imipenem/cilastatin, meropenem, and doripenem) and antibiotics against MRSA (vancomycin, teicoplanin, daptomycin, and linezolid).

**Phase 2: Establishing an ID consultation service and implementing the ASP (April 1, 2020, to March 31, 2021).**   The ID consultation and ASP were introduced on April 1, 2020. The ID consultation service is a system wherein for 5 days per week, full-time ID physicians provide ID consultations for referrals from the other 15 departments, review positive blood culture results to ensure that patients receive the appropriate empirical treatment, and provide feedback to physicians. The ASP was structured as follows: (1) for 3 days a week, the AST evaluated the medical records and antimicrobial use in patients who were administered specific broad-spectrum antibiotics (vancomycin, teicoplanin, daptomycin, linezolid, cefepime, cefozopran, piperacillin–tazobactam, imipenem–cilastatin, meropenem, and doripenem) and provided post-prescription review with feedback to the physician; (2) patients who were difficult to assess during AST rounds alone were directly evaluated by the ID physician at the bedside; (3) the AST performed an intervention evaluation using prespecified categories (Table 1), based on culture results, treatment course, and treatment duration. The assessments were conducted within 48 h of the initial intervention or within the period indicated by the AST (e.g., de-escalation when culture results were available). The evaluations were classified based on the consensus of all AST members. Discordance between the AST recommendations and the clinician's opinion, if any, were resolved through discussion, and the final decision was based on the clinician's judgment. At the time of the evaluation, patients with ID consultations were excluded from the AST evaluation.

## Primary outcome measures

The primary outcome measured was the change in days of therapy with carbapenem (CAR-DOT; for imipenem–cilastatin, meropenem, and doripenem), expressed as DOT per 100 patient-days per month.

**Table 1. Assessment sheet used by the AST.**

| | | Category | |
|---|---|---|---|
| Appropriate therapy | A | Appropriate | Antimicrobial selection and dosage are appropriate at the time of evaluation. |
| | B | Better choice | There are no major problems with antimicrobials selection, although there are suggestions for some modifications and changes. |
| Inappropriate therapy | C | Culture | Absence or inadequacy of submission of bacterial cultures; requires additional investigation (or, therefore, is difficult to evaluate) |
| | D | De-escalation | Broad-spectrum antimicrobials were used based on the clinical characteristics, culture results, and local factors, and these can be changed to a narrow-spectrum antimicrobial. |
| | E | Escalation | The antimicrobials do not provide adequate coverage of the target microorganisms; therefore, the spectrum needs to be broadened or the antimicrobial should be changed. |
| | F | Fitting dose | The dose and method of administering the antimicrobials are inappropriate due to the renal function or other factors; thus, adjustments are necessary. |
| | H | Halt | Discontinuation is necessary because the purpose of antimicrobial administration has been achieved, further use is unnecessary, or there is a risk of allergy. |
| | I | Indication document | The purpose of use and the target microorganisms of the antimicrobials are not described in the medical record and, therefore, cannot be evaluated; additional descriptions are needed. |
| Time out | T | Time out | Notify physician that culture results are available (3–5 days after initiation of the antimicrobial therapy) or that it is time to reconsider the termination of antimicrobial therapy (10–14 days after initiation of the antimicrobial therapy). |

Abbreviation: AST, Antimicrobial Stewardship Team.

## Secondary outcome measures

**DOT for antipseudomonal agents.**   The total DOT per month per 100 patient-days was calculated for three antipseudomonal agents: piperacillin–tazobactam, cefepime, and cefozopran. This was because the aforementioned antibiotics are broad-spectrum antimicrobials, similar to CAR. The DOT for antipseudomonal agents was evaluated to assess whether CAR was simply being replaced with other broad-spectrum antipseudomonal agents.

**DOT for ampicillin, ampicillin–sulbactam, cefazolin, and cefmetazole.**   The total DOT per month per 100 patient-days was calculated for four antimicrobial agents, namely ampicillin, ampicillin–sulbactam, cefazolin, and cefmetazole, as these antibiotics are narrow-spectrum antimicrobials after de-escalation. This parameter was assessed to confirm whether the intervention resulted in appropriate de-escalation.

**Antimicrobial Use Density (AUD) for carbapenem, three antipseudomonal agents, four narrow-spectrum antimicrobials.**   AUD was also calculated to normalize the amount of each antimicrobial consumption. AUD was expressed as defined daily dose (DDD) per 100 patient-days for individual antimicrobial agent (AUD = antimicrobial consumption/DDD×100/patient-days). DDD per 100 inpatient-days for each drug or drug category prescribed monthly was calculated following the 2021 World Health Organization (WHO) Anatomical Therapeutic Chemical (ATC) classification system [14]. AUDs were calculated for carbapenem (CAR-AUD; for imipenem–cilastatin, meropenem, and doripenem); three antipseudomonal agents (piperacillin–tazobactam, cefepime, and cefozopran); and four narrow-spectrum antimicrobials (ampicillin, ampicillin–sulbactam, cefazolin, and cefmetazole).

**Incidence of hospital-acquired resistant microorganisms/*Clostridioides difficile* infection (CDI), and candidemia.**   We measured the annual incidence of antibiotic-resistant bacterial infections, CDI, and candidemia per 1000 patient-days between April 2018 and March 2021, the period for which these study data were available, as an indicator of the outcome of the ASP. The resistant microorganisms included CPE, CRE, multidrug-resistant *P. aeruginosa* (MRPA), MRSA, and extended-spectrum beta-lactamase (ESBL)-producing *Enterobacteriaceae*. Hospital-acquired microorganisms were defined as those that were identified >72 h after

admission [15]. To exclude duplicates, when the same resistant microorganisms were detected more than once in the same patient, only the first specimen obtained in each month was included in the analysis [15]. However, if the resistant microorganism was detected in blood samples, this infection was defined as a new event if the same resistant bacteria had not been detected in blood samples from the same patient within the past 2 weeks [15]. Only clinical specimens of resistant microorganisms were included, and specimens for surveillance culture and negative confirmation were excluded. CDI was defined as the number of patients with positivity of CD toxin (C. DIFF QUIK CHEK COMPLETE; Alere Medical Co., Tokyo, Japan). Moreover, ESBL-producing *Enterobacteriaceae*, including *E. coli*, *Klebsiella pneumoniae*, *Klebsiella oxytoca*, and *Proteus mirabilis* were identified using the Cica β-test (Kanto Chemical Co., Tokyo, Japan) and disc diffusion method. CPE were identified using the modified carbapenem inactivation method, which was performed according to CLSI M100-S27 [16]. CRE was defined as either resistance to meropenem [minimum inhibitory concentrations (MIC) $\geq 2$ μg/mL] or resistance to both imipenem (MIC $\geq 2$ μg/mL) and cefmetazole (MIC $\geq 64$ μg/mL) [17]. MRPA was defined as *P. aeruginosa* that is resistant to CAR, aminoglycosides, and fluoroquinolones according to the standard set by the Clinical and Laboratory Standards Institute's for antimicrobial susceptibility tests, regardless of whether carbapenemase was produced or not [12]. Furthermore, we included the incidence of candidemia, although the blood samples intended for confirming negative candidemia were excluded from this analysis.

**Total number of inpatient specimens.**   To assess the impact of ID consultation and application of the ASP, we evaluated the total number of inpatient samples from April 2018 to March 2021.

**Cost of CAR and all intravenous antimicrobials.**   To assess the economic impact of ASPs, we assessed the cost of purchasing antimicrobials per patient-day each year from April 2018 to March 2021. The adjusted purchase costs were calculated based on the actual purchase costs (considering the cost of switching from branded to generic products and changes in drug prices) and prices of branded agents in April 2020. The exchange rate of 1 USD to 108 JPY was used for the calculations in March 2021.

**Assessment and acceptance rates of proposals by the AST.**   The evaluation (Table 1) and acceptance rates by the AST were calculated. Acceptance rate was the sum of the accepted and partially accepted suggestions divided by the total number of suggestions.

**All-cause in-hospital mortality and length of hospital stay.**   Data for all-cause in-hospital mortality and length of hospital stay were extracted and included in the analysis.

## Statistical analysis

To demonstrate the effect of each intervention on CAR-DOT, CAR-AUD, the DOT, and AUD for the three antipseudomonal agents (piperacillin–tazobactam, cefepime, and cefozopran) and four narrow-spectrum antibiotics (ampicillin, ampicillin–sulbactam, cefazolin, and cefmetazole) were ascertained. Furthermore, we carried out a segmented regression analysis of interrupted time-series studies for the two periods (Phase 1: April 1, 2018 to March 31, 2020; Phase 2: April 1, 2020 to March 31, 2021), when the DOT data were available. Trends and changes in the levels of the incidence of antibiotic-resistant bacterial infection, CDI, candidemia, mortality, and length of hospital stay were evaluated using a segmented regression analysis of interrupted time-series studies for the same periods. We adopted the most standard model based on the linear regression model, including level change and slope change. Autocorrelation (confirmed by the p-value of the Ljung-Box test and visualization of autocorrelation) was observed in the DOT for the antipseudomonal agents, incidence of CRE, MRPA, MRSA, candidemia, in-hospital mortality, and length of hospital stay. However, we did not adjust for

autocorrelation since the number of time points was small (36 before and after the intervention) and we wanted to discuss the same model consistently for the primary outcome without autocorrelation issues. Therefore, this study was set as an exploratory (hypothesis-generating) research. A bivariate analysis for the cost of antimicrobials and the total number of inpatient specimens was carried out using the Mann–Whitney U test (continuous variables), with p<0.05 regarded as statistically significant. R software, version 4.0.2. (The R Foundation for Statistical Computing, Vienna, Austria) was used for all analyses.

### Ethical considerations

This study was approved by the Institutional Review Board of the ACC hospital (approval number: 2020-1-640) and conducted according to the principles of the Declaration of Helsinki. The requirement of informed consent was waived because this study only used data that were collected in clinical practice.

## Results

During the study period, 32,202 patients were admitted to the ACC, with 22,096 and 10,106 in the baseline and intervention periods, respectively. During the intervention period, the AST provided a total of 913 instances of feedback regarding the specific broad-spectrum antimicrobials, and there were 440 ID consultations.

### Trends in CAR use

Fig 1 shows the changes in the CAR-DOT during the two phases. After initiation of the intervention, the trend in the monthly CAR-DOT decreased (coefficients: −0.14; 95% confidence interval [CI]: −0.22 to −0.06, p = 0.001) and the level in the monthly CAR-DOT also reduced (coefficients: −1.16; 95% CI: −1.74 to −0.55, p<0.001). The trend in the CAR-AUD decreased

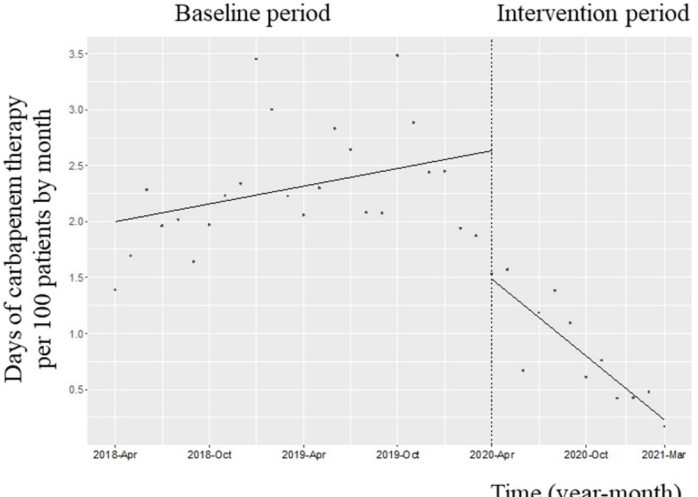

**Fig 1. Trends in the days of carbapenem therapy per 100 patients, by month, during Phase 2 of the intervention period.** Each dot refers to the days of carbapenem therapy per 100 patients in each month, and the slope is based on the linear regression in the two phases. The explanation of each phase is as follows: Phase 1 (antimicrobial notification by the infection control team from April 1, 2018, to March 31, 2020); Phase 2 (establishing an infectious disease [ID] consultation service and implementation of the Antimicrobial Stewardship Program [ASP] from April 1, 2020, to March 31, 2021). The trend in the monthly CAT-DOT decreased (coefficients: −0.14; 95% confidence interval [CI]: −0.22 to −0.06, p = 0.001), and its levels reduced (coefficients: −1.16; 95% CI: −1.74 to −0.55, p<0.001).

along with its level (trend change, coefficient: -0.10; 95% CI: -0.15 to 0.05, p<0.001; change in level, coefficient: -0.60; 95% CI: -0.98 to -0.23, p<0.001).

## Use of antipseudomonal agents

Fig 2 shows the changes in the monthly DOT of the three antipseudomonal agents (piperacillin–tazobactam, cefepime, and cefozopran). However, the level of the monthly DOT for the three antipseudomonal agents did not decrease (coefficient: 0.20; 95% CI: −0.67 to 1.09, p = 0.65), although its trend reduced (coefficient: −2.22; 95% CI: −0.33 to −0.10, p<0.001). The level of the monthly AUD for the three antipseudomonal agents did not decrease (coefficient: -0.01; 95% CI: -0.75 to 0.73, p = 0.98), although its trend reduced (coefficient: -0.12; 95% CI: -0.22 to -0.031, p = 0.01).

## Use of narrow-spectrum antibiotics

Fig 3 shows the changes in the monthly DOT of the four narrow-spectrum antibiotics (ampicillin, ampicillin–sulbactam, cefazolin, and cefmetazole). The trend in the monthly DOT of the four narrow-spectrum antibiotics increased along with its level (trend change, coefficient: 0.47; 95% CI: 0.37 to 0.57, p<0.001; change in level, coefficient: 1.76; 95% CI: 1.00 to 2.53, p<0.001). The trend in the monthly AUD of the four narrow-spectrum antibiotics increased along with its level (trend change, coefficient: 0.43; 95% CI: 0.36 to 0.51, p<0.001; change in level, coefficient: 1.45; 95% CI: 0.85 to 2.04, p<0.001).

## Incidence of antimicrobial-resistant microorganisms, CDI, and candidemia

During the study period, CPE was not detected. The change in the level of the monthly incidence of MRPA did not decrease (coefficient: 0.09; 95% CI: −0.015 to 0.19, p = 0.10); however,

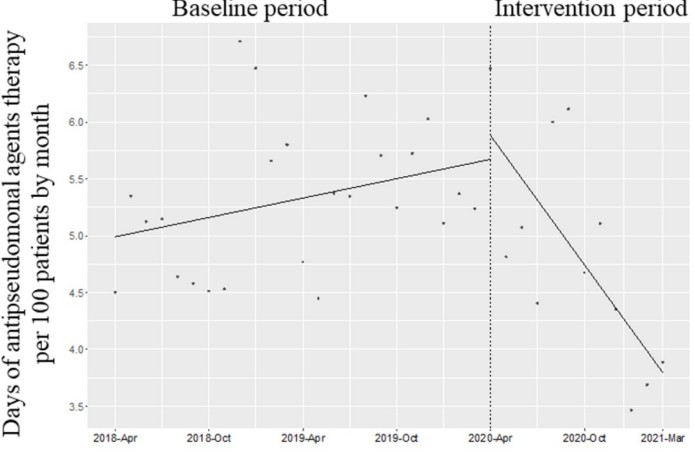

**Fig 2. Trends in the days of antipseudomonal agent (piperacillin–tazobactam, cefepime, and cefozopran) therapy per 100 patients, by month, during Phase 2 of the intervention period.** Each dot refers to the days of antipseudomonal agent therapy per 100 patients in each month, and the slope is based on linear regression in the two phases. The explanation of each phase is as follows: Phase 1 (antimicrobial notification by the infection control team from April 1, 2018, to March 31, 2020); Phase 2 (establishing an infectious disease [ID] consultation service and implementation of the Antimicrobial Stewardship Program [ASP] from April 1, 2020, to March 31, 2021). The level of the monthly DOT of the three antipseudomonal agents did not decrease (coefficient: 0.20; 95% confidence interval [CI]: −0.67 to 1.09, p = 0.65), although the trend decreased (coefficient: −2.22; 95% CI: −0.33 to −0.10, p<0.001).

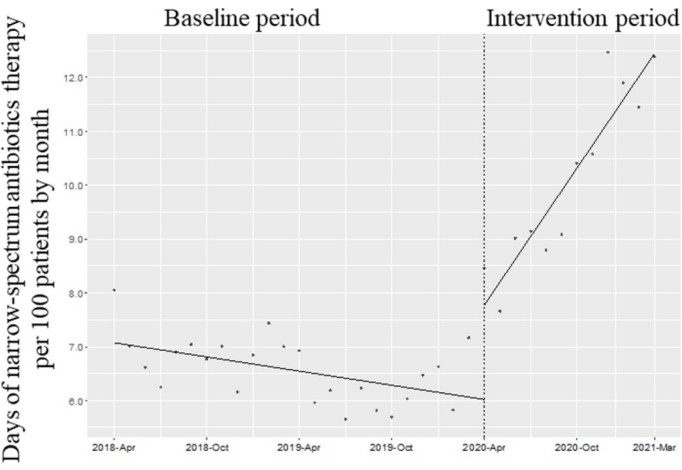

**Fig 3. Trends in the days of narrow-spectrum antibiotic (ampicillin, ampicillin–sulbactam, cefazolin, and cefmetazole) therapy per 100 patients, by month, during Phase 2 of the intervention period.** Each dot refers to the days of narrow-spectrum antibiotic therapy per 100 patients in each month, and the slope is based on linear regression in the two phases. The explanation of each phase is as follows: Phase 1 (antimicrobial notification by the infection control team from April 1, 2018, to March 31, 2020); Phase 2 (establishing an infectious disease [ID] consultation service and implementation of the Antimicrobial Stewardship Program [ASP] from April 1, 2020, to March 31, 2021). The trend in the monthly DOT of the four narrow-spectrum antibiotics increased (coefficient: 0.47; 95% confidence interval [CI]: 0.37 to 0.57, p<0.001), and its level increased (coefficient: 1.76; 95% CI: 1.00 to 2.53, p<0.001).

there was a significant reduction in the MRPA trend (coefficient: −0.02; 95% CI: −0.02 to −0.004, $p$ = 0.02); Fig 4). The level of the monthly incidence of MRSA did not decrease

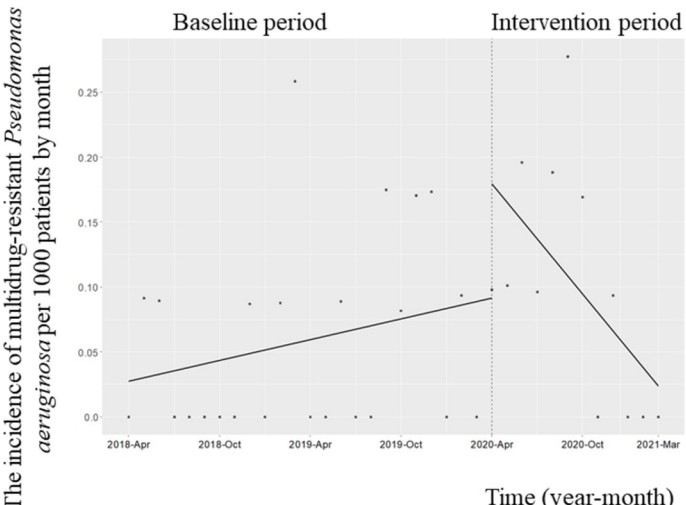

**Fig 4. Trends in the incidence of multidrug-resistant *Pseudomonas aeruginosa* (MRPA) per 1000 patients, by month, during Phase 2 of the intervention period.** Each dot refers to the incidence of MRPA per 1000 patients each month and the slope is based on linear regression in the two phases. The explanation of each phase is as follows: Phase 1 (antimicrobial notification by the infection control team from April 1, 2018, to March 31, 2020); Phase 2 (establishing an infectious disease [ID] consultation service and implementation of the Antimicrobial Stewardship Program [ASP] from April 1, 2020, to March 31, 2021). The number of isolated samples in 2018, 2019, and 2020 was 7, 9, and 13, respectively. The level of the monthly incidence of MRPA did not decrease (coefficient: 0.09; 95% confidence interval [CI]: −0.015 to 0.19, $p$ = 0.10), although the trend of the infection was significantly reduced (coefficient: −0.02; 95% CI: −0.02 to −0.004, $p$ = 0.02).

(coefficient: 0.12; 95% CI: −0.16 to 0.42, $p$ = 0.40), although the trend was significantly reduced (coefficient: −0.05; 95% CI: −0.09 to −0.013, $p$ = 0.01; Fig 5). There was no significant change in the trend of the monthly incidence (coefficient: 0.012; 95% CI: −0.001 to 0.03, $p$ = 0.09) and the associated change in the level after the implementation of the ASP (coefficient: −0.02; 95% CI: −0.12 to 0.09, $p$ = 0.78; S1 Fig). There was no significant change in the trend of the monthly incidence of ESBL-producing *Enterobacteriaceae* (coefficient: 0.018; 95% CI: −0.001 to 0.04, $p$ = 0.07) or its level (coefficient: −0.03; 95% CI: −0.18 to 0.12, $p$ = 0.67; S2 Fig). The trend and change in the level of the monthly incidence of CDI did not decrease significantly (trend change, coefficient: −0.004; 95% CI: −0.02 to 0.013, $p$ = 0.68; change in level, coefficient: 0.01; 95% CI: 0.02 to 0.24, p = 0.12; S3 Fig). The trend of the monthly incidence of candidemia (coefficient: −0.004; 95% confidence interval [CI]: −0.02 to 0.013, $p$ = 0.45) and the change in its level (coefficient: −0.01; 95% CI: −0.02 to 0.24, $p$ = 0.82; S4 Fig) did not decrease significantly.

## Total number of inpatient specimens

Regarding the total number of patients, the number of samples increased significantly after the intervention (median: 574.5 vs. 669.5, $p<0.001$).

## Cost of CAR and all intravenous antimicrobials

Table 2 shows the actual and adjusted CAR purchase costs for all intravenous antimicrobials in each year. The actual CAR purchase cost per patient-days significantly decreased after the initiation of Phase 2 of this program ($p<0.001$). However, there was no significant change in the actual cost per patient-days of all intravenous antimicrobials that were purchased ($p$ = 0.83). The adjusted CAR purchase cost per patient-days significantly decreased after the

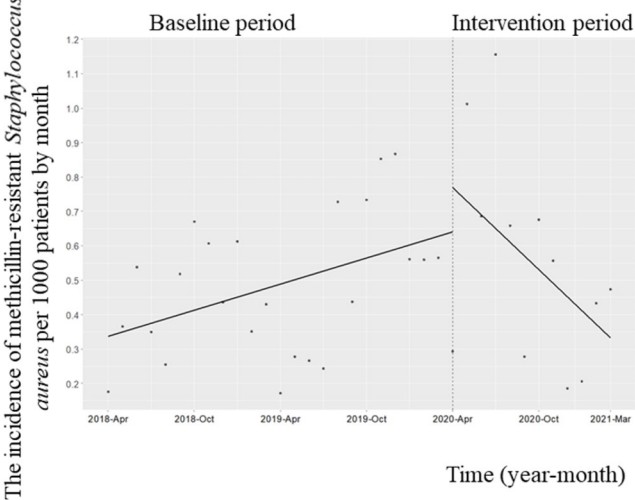

**Fig 5. Trends in the incidence of methicillin-resistant *Staphylococcus aureus* (MRSA) per 1000 patients, by month, during Phase 2 of the intervention period.** Each dot refers to the incidence of MRSA per 1000 patients each month and the slope is based on linear regression in the two phases. The explanation of each phase is as follows: Phase 1 (antimicrobial notification by the infection control team from April 1, 2018, to March 31, 2020); Phase 2 (establishing an infectious disease [ID] consultation service and implementation of the Antimicrobial Stewardship Program [ASP] from April 1, 2020, to March 31, 2021). The number of isolated samples in 2018, 2019, and 2020 was 61, 72, and 69, respectively. The level of the monthly incidence of MRSA did not decrease (coefficient: 0.12; 95% confidence interval [CI]: −0.16 to 0.42, p = 0.40), although there was a significant reduction in the trend of the infection (coefficient: −0.05; 95% CI: −0.09 to −0.013, p = 0.01).

**Table 2. Purchase costs of carbapenems and all intravenous antimicrobials per patient-days from 2018 to 2020.**

| Year | Actual cost of CAR, USD | Adjusted cost of CAR, USD | Actual cost of all intravenous antimicrobials, USD | Adjusted cost of intravenous antimicrobials, USD |
|------|------|------|------|------|
| 2018 | 5.38 | 7.93 | 34.36 | 52.04 |
| 2019 | 6.01 | 9.33 | 36.97 | 58.20 |
| 2020 | 2.28 | 3.73 | 35.33 | 56.40 |

The actual cost includes the cost of switching to generic drugs and factors in the changes in the drug prices.

The adjusted cost is calculated based on the drug price in April 2020.

initiation of Phase 2 ($p < 0.001$). Nonetheless, there was no significant change in the adjusted purchase cost per patient-days for all intravenous antimicrobials ($p = 0.57$).

## Assessment and acceptance rates of proposals by the AST

Based on the evaluation by the AST, we determined that there were 347 appropriate (38%) and 566 inappropriate (62%) instances of antimicrobial use (Table 3). "De-escalation" was the commonest type of inappropriate use (42%). The overall acceptance rate of the AST suggestions was 76%.

## All-cause in-hospital mortality and length of hospital stay

There was no significant change in the trend of in-hospital mortality (coefficient: −0.10; 95% CI: −0.23 to 0.02, $p = 0.122$) and length of hospital stay (coefficient: −0.06; 95% CI: −0.13 to 0.01, $p = 0.12$), or in their level (coefficient: −0.36; 95% CI: −1.33 to 0.60, $p = 0.47$ and coefficient: −0.49; 95% CI: −1.1 to 0.08, $p = 0.10$, S5 and S6 Figs, respectively).

## Discussion

This is the first study to report on the effects of ID consultation and ASP interventions in a cancer center in Japan. Cancer centers constitute challenging facilities for conducting antibiotic stewardship interventions due to the complexity of patients conditions, on account of the underlying cancers and hematologic malignancies, neutropenia, the various departments that are involved in the multidisciplinary patients' management, and the varying departmental

**Table 3. Content and acceptance rate of feedback by the AST with regard to specific broad-spectrum antimicrobial usage.**

| Category | Number of evaluations (n) | Suggestions (n) | Accepted (n) | Partially accepted (n) | Not accepted (n) | Acceptance rate[a] (%) |
|------|------|------|------|------|------|------|
| A | 251 | 54 | 38 | 4 | 12 | 78% |
| B | 96 | 79 | 41 | 8 | 30 | 62% |
| C | 89 | 54 | 29 | 4 | 21 | 61% |
| D | 235 | 232 | 176 | 12 | 44 | 81% |
| E | 62 | 61 | 60 | 0 | 1 | 98% |
| F | 36 | 35 | 24 | 2 | 9 | 74% |
| H | 91 | 88 | 58 | 4 | 26 | 70% |
| I | 51 | 34 | 22 | 4 | 8 | 76% |
| T | 2 | 1 | 1 | 0 | 0 | 100% |
| Total | 913 | 638 | 449 | 38 | 151 | 76% |

Abbreviations: A, Appropriate; B, Better choice; C, Culture; D, De-escalation; E, Escalation; F, Fitting dose; H, Halt; I, Indication document; T, Time out.

[a] Acceptance rate is the sum of accepted and partially accepted suggestions divided by the total number of suggestions.

structures and internal guidelines [7]. However, our intervention was effective in reducing CAR consumption without increasing the consumption of alternative broad-spectrum antimicrobials or resulting in other negative outcomes.

In our study, the level and trend of CAR-DOT was significantly decreased, and only the trend of the DOT for the three antipseudomonal agents was significantly decreased. The levels of the three antipseudomonal agents did not decrease significantly, but the results suggest that a decrease can be expected in the future if significant decrease in trend is sustained. There was no significant decrease in the level of the three antipseudomonal agents, suggesting that a decrease might be expected with long-term observation. This finding might have been due to the routine use of aggressive chemotherapy with febrile neutropenia in cancer centers [18]. This study also evaluated monthly DOT trends for the narrow-spectrum antimicrobials to confirm de-escalation from broad-spectrum antimicrobial usage. The use of these drugs has not been previously investigated [8, 12]. In our study, the level and trend of CAR-DOT and the trend of DOT for antipseudomonal agents was decreased, whereas the level and trend of the narrow-spectrum antimicrobials was increased. The results of the analysis between antimicrobial use using AUD and DOT were consistent. Although there were no significant differences, the length of hospital stay and mortality rates tended to decrease and are expected to decrease with long-term observation. These results of no increase in mortality and hospital stay, suggest that antimicrobial agents were being used appropriately. For more accurate assessment in future, it is thought that mortality due to ID should also be evaluated, but individual patient's data were not evaluated in this study. Discontinuation or de-escalation of antimicrobial agents in patients with malignancy is often avoided [4, 8]; however, with ID consultation and ASP, they can be implemented safely without negative patient outcomes.

The main concept of ASPs in any patient population, including cancer patients, is to facilitate antimicrobial use such that each patient receives the most effective and safest antimicrobials for treating their infection while minimizing the ecological impact of the antimicrobials used [1]. Given the high frequency of infection in cancer patients and the need for both therapeutic and prophylactic use of antimicrobials, the magnitude of antimicrobial usage is substantial. This imposes a significant antimicrobial pressure not only on the patient's normal microbiota, but also potentially on the surrounding environment. An indirect consequence of increased antimicrobial use, for example, is a higher prevalence of CDI in this population [19]. Moreover, broad-spectrum antimicrobial use increases the risk for candidemia [20]. Furthermore, the greatest concern associated with increased antimicrobial usage is the emergence of infections caused by drug-resistant bacteria [7]. In our study, there was a significant decrease in the trend of MRPA and MRSA, although there was no change in the number of infections caused by other drug-resistant bacteria. There was no significant difference between candidemia and CDI; however, a decreasing trend was observed. This might have been due to the short observation period of our study, which implied that a marked decrease can be expected with long-term observation in future evaluations [8, 13]. In particular, as the trend of MRPA and MRSA incidence was significantly reduced in our study, additional observation for a short-term period is expected to further reduce the level of MRPA and MRSA incidence. Moreover, the total number of inpatient specimens were significantly increased during the study period. Although there were no significant differences between CRE and ESBL-producing *Enterobacteriaceae*, an increasing trend was observed, which might have been influenced by an increase in the total number of samples. Especially in Japan, the incidence of cefotaxime resistant *E. coli* and *K. pneumoniae*, which are suspected to be ESBL-producing, has been increasing every year [2]. No in-hospital outbreaks of CRE or ESBL-producing Enterobacteriaceae were observed during the study period. Therefore, the detection rate of resistant bacteria

that was possibly underestimated before the intervention might have edged closer to the true detection rate, and this factor might have affected our results.

Our interventions began in April 2020, but changes have been occurring in the use of carbapenems and the three anti-pseudomonal agents since late 2019. This was partly due to the impact of a one-hour lecture on the appropriate use of antimicrobials to healthcare providers by an external ID expert on February 19, 2019.

In our study, the total number of samples was increased after the intervention. This is an important finding for the promotion of diagnostic stewardship. Diagnostic stewardship refers to the performing of the appropriate test on the appropriate patient at the appropriate time, to provide optimal clinical care [21]; it is an integral part of ASP [22]. At the beginning of our intervention, the ASP was impeded by the insufficient number of specimens submitted; however, as the total number of specimens increased, the ASP could be conducted appropriately. Conversely, without active ASP intervention, the diagnostic stewardship program may be ineffective [23], and a strategy that combines both is needed to optimize the care of patients with cancer.

Our results showed that the decrease in CAR-DOT due to ID consultation and ASP intervention led to a significant reduction in the cost of CAR. While the use of alternative antimicrobials was increased, the overall cost of antimicrobials did not change significantly after adjustment, indicating that our intervention did not impose an economic burden on the hospital.

The acceptance rate for our ASP was as high as 76% in this study. Other studies have reported similarly high acceptance rates of ASP (76–87%) [8, 13]. The high compliance rate suggests that the results of our study are likely positive. Moreover, the high compliance rate could be attributed to the approval by the hospital director who informed all the institution's medical staff about the importance of the intervention and about antimicrobial resistance, before the intervention started. Additionally, this may be because the ASP and ID consultations resulted in appropriate interventions and management of cases with difficult diagnoses and treatments. Most cancer centers in Japan do not have cross-organ departments such as the Department of General Internal Medicine; the Department of ID plays a part in this role [5]. It is possible that these activities increased the satisfaction of the oncologists; consequently, they became more compliant with ASP.

There is scarcity of ID physicians in Japan. Ideally, Japan needs 3,000–4,000 ID physicians; however, as of August 2021, there were only 1622 ID physicians (i.e., 1.29 per 100,000 population in Japan vs. 2.78 per 100,000 in the United States) [24–26]. Furthermore, the training of ID physicians, pharmacists, clinical laboratory technicians, and infection control nurses in cancer centers is often time-consuming, since the complex background of patients with cancer requires sufficient experience and knowledge of ID and oncology. Therefore, the lack of appropriate human resources at specialized facilities such as cancer centers is one of the major barriers to implementing ASP. However, our study showed a positive effect of the ASP with the involvement of fewer human resources, which suggests that in cancer centers in Japan, the ASP would be a highly effective activity that can be sufficiently effective even when few specialists are involved.

This study has several limitations. First, as this was a single-center study at a Japanese cancer center, it is unclear whether the hospital-based ASP strategy is generalizable. Second, the impact of broad-spectrum antimicrobial prescribing for patients with coronavirus disease (COVID-19) and for hospitalized patients suspected to have COVID-19 during the COVID-19 pandemic needs to be considered [27], since broad-spectrum antimicrobials have been used both prophylactically and therapeutically for such patients [27, 28]. Although our institution was affected by the COVID-19 pandemic, the trends of broad-spectrum antimicrobial

prescribing declined. Third, in 2019, a nationwide cefazolin shortage occurred in Japan due to manufacturing-related problems at a major production facility [29]. Although the cefazolin shortage led to a significant increase in the use of alternative antimicrobials at several facilities in Japan [29], our data trends from 2018 to 2019 showed a decline in broad-spectrum antimicrobial use in 2020. Additionally, there was a supply restriction of cefepime from June 2018 to October 2019; nonetheless, broad-spectrum antimicrobial use declined in 2020 compared to the usage trends in 2018 and 2019. To resolve these limitations, a long-term multicenter study including other Japanese cancer centers is warranted in the future.

In conclusion, the combination of ID consultation and ASP in this study reduced the use of CAR without negative impacts on the outcome-related parameters, such as mortality rates and length of hospital stay. The introduction of these systems may lead to the establishment of safer cancer treatment and improve the overall prognosis for patients with cancer in Japan.

## Supporting information

**S1 Fig. Trends of the incidence of carbapenem-resistant *Enterobacteriaceae* (CRE) per 1000 patients, by month, during Phase 2 of the intervention period.** Each dot refers to the incidence of CRE per 1000 patients each month and the slope is based on linear regression in the two phases. The explanation of each phase is as follows: Phase 1 (antimicrobial notification by the infection control team from April 1, 2018, to March 31, 2020); Phase 2 (establishing an infectious disease [ID] consultation service and implementation of the Antimicrobial Stewardship Program [ASP] from April 1, 2020, to March 31, 2021). The number of isolated samples in 2018, 2019, and 2020 was 10, 5, and 8, respectively. There was no increase in the trend of the monthly incidence of CRE (coefficient: 0.012; 95% confidence interval [CI]: −0.001 to 0.03, $p = 0.09$) or a change in its level after the implementation of the Antimicrobial Stewardship Program (coefficient: −0.02; 95% CI: −0.12 to 0.09, $p = 0.78$).
(DOC)

**S2 Fig. Trends of the incidence of extended-spectrum beta-lactamase (ESBL)-producing *Enterobacteriaceae* per 1000 patients, by month, during Phase 2 of the intervention period.** Each dot refers to the incidence of ESBL-producing Enterobacterales per 1000 patients each month and a slope based on linear regression in the two phases. The explanation of each phase is as follows: Phase 1 (antimicrobial notification by the infection control team from April 1, 2018, to March 31, 2020); Phase 2 (establishing an infectious disease [ID] consultation service and implementation of the Antimicrobial Stewardship Program [ASP] from April 1, 2020, to March 31, 2021). The number of isolated samples in 2018, 2019, and 2020 was 20, 24, and 39, respectively. There was no significant increase in the trend of the monthly incidence of ESBL-producing Enterobacteriaceae (coefficient: 0.018; 95% confidence interval [CI]: −0.001 to 0.04, $p = 0.07$) and the change in its level (coefficient: −0.03; 95% CI: −0.18 to 0.12, $p = 0.67$).
(DOC)

**S3 Fig. Trends of the incidence of *Clostridioides difficile* infection (CDI) per 1000 patients, by month, during Phase 2 of the intervention period.** Each dot refers to the incidence of CDI per 1000 patients each month and the slope is based on linear regression in the two phases. The explanation of each phase is as follows: Phase 1 (antimicrobial notification by the infection control team from April 1, 2018, to March 31, 2020); Phase 2 (establishing an infectious disease [ID] consultation service and implementation of the Antimicrobial Stewardship Program [ASP] from April 1, 2020, to March 31, 2021). The number of isolated samples in 2018, 2019, and 2020 was 8, 12, and 20, respectively. There was no significant reduction in the trend or a change in the level of the monthly incidence of CDI (trend change, coefficient:

−0.004; 95% confidence interval [CI]: −0.02 to 0.013, $p = 0.68$; change in level, coefficient: 0.01; 95% CI: 0.02 to 0.24, $p = 0.12$).
(DOC)

**S4 Fig. Trends of the incidence of candidemia per 1000 patients, by month, during Phase 2 of the intervention period.** Each dot refers to the incidence of candidemia per 1000 patients each month and the slope is based on linear regression in the two phases. The explanation of each phase is as follows: Phase 1 (antimicrobial notification by the infection control team from April 1, 2018, to March 31, 2020); Phase 2 (establishing an infectious disease [ID] consultation service and implementation of the Antimicrobial Stewardship Program [ASP] from April 1, 2020, to March 31, 2021). The number of isolated samples in 2018, 2019, and 2020 was 9, 8, and 3, respectively. There was no significant reduction in the trend of the monthly incidence of candidemia (coefficient: −0.004; 95% confidence interval [CI]: −0.02 to 0.013, $p = 0.45$) or a change in its level (coefficient: −0.01; 95% CI: −0.02 to 0.24, $p = 0.82$).
(DOC)

**S5 Fig. Trends of the in-hospital mortality by month during Phase 2 of the intervention period.** Each dot refers to the in-hospital mortality each month and the slope is based on linear regression in the two phases. The explanation of each phase is as follows: Phase 1 (antimicrobial notification by the infection control team from April 1, 2018, to March 31, 2020); Phase 2 (establishing an infectious disease [ID] consultation service and implementation of the Antimicrobial Stewardship Program [ASP] from April 1, 2020, to March 31, 2021). There was no significant change in the trend of in-hospital mortality (coefficient: −0.10; 95% confidence interval [CI]: −0.23 to 0.02, $p = 0.122$) or in its level (coefficient: −0.36; 95% CI: −1.33 to 0.60, $p = 0.47$).
(DOC)

**S6 Fig. Trends of the length of hospital stay by month during Phase 2 of the intervention period.** Each dot refers to the length of hospital stay each month and the slope is based on linear regression in the two phases. The explanation of each phase is as follows: Phase 1 (antimicrobial notification by the infection control team from April 1, 2018, to March 31, 2020); Phase 2 (establishing an infectious disease [ID] consultation service and implementation of the Antimicrobial Stewardship Program [ASP] from April 1, 2020, to March 31, 2021). There was no significant change in the trend of the length of hospital stay (coefficient: −0.06; 95% confidence interval [CI]: −0.13 to 0.01, $p = 0.12$) or its level (coefficient: −0.49; 95% CI: −1.1 to 0.08, $p = 0.10$).
(DOC)

**S1 Dataset.**
(CSV)

## Acknowledgments

We are grateful to all the clinical staff of Aichi Cancer Center Hospital for their commitment toward providing patient care and to the staff of the Department of Infectious Diseases, Okinawa Prefectural Chubu Hospital, for providing the AST evaluation sheet (Table 1).

## Author Contributions

**Conceptualization:** Naoya Itoh, Nana Akazawa.

**Data curation:** Naoya Itoh, Nana Akazawa, Eri Kanawaku, Hiromi Murakami, Yuichi Ishibana, Daichi Kawamura.

**Formal analysis:** Takanori Kawabata, Keita Mori.

**Investigation:** Naoya Itoh.

**Methodology:** Naoya Itoh.

**Project administration:** Naoya Itoh.

**Resources:** Naoya Itoh.

**Supervision:** Eiichi N. Kodama, Norio Ohmagari.

**Validation:** Naoya Itoh.

**Visualization:** Naoya Itoh.

**Writing – original draft:** Naoya Itoh.

**Writing – review & editing:** Naoya Itoh, Nana Akazawa, Eri Kanawaku, Hiromi Murakami, Yuichi Ishibana, Daichi Kawamura, Takanori Kawabata, Keita Mori, Eiichi N. Kodama, Norio Ohmagari.

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
