## [Decision Letter · Decision Letter 0]

4 Nov 2021

PONE-D-21-27727Impact of infectious disease consultation and antimicrobial stewardship program at a Japanese cancer center: an interrupted time-series analysisPLOS ONE

Dear Dr. Itoh,

Thank you for submitting your manuscript to PLOS ONE. After careful consideration, we feel that it has merit but does not fully meet PLOS ONE’s publication criteria as it currently stands. Therefore, we invite you to submit a revised version of the manuscript that addresses the points raised during the review process.

We look forward to receiving your revised manuscript.

Kind regards,

Hiroshi Nishiura

Academic Editor

PLOS ONE

Journal Requirements:

a) Did participants provide their written or verbal informed consent to participate in this study?

b) If consent was verbal, please explain i) why written consent was not obtained, ii) how you documented participant consent, and iii) whether the ethics committees/IRB approved this consent procedure

Additional Editor Comments (if provided):

One reviewer gave comment on the use of DOT and raised concern over the interpretation of ITS. The other reviewer gave comments for minor technical improvement. With little additional work, your study will be polished and in a good shape for publication as an original research article.

Reviewers' comments:

Reviewer's Responses to Questions

**Comments to the Author**

1. Is the manuscript technically sound, and do the data support the conclusions?

Reviewer #1: Yes

Reviewer #2: Yes

2. Has the statistical analysis been performed appropriately and rigorously? 

Reviewer #1: I Don't Know

Reviewer #2: Yes

3. Have the authors made all data underlying the findings in their manuscript fully available?

Reviewer #1: No

Reviewer #2: Yes

4. Is the manuscript presented in an intelligible fashion and written in standard English?

Reviewer #1: Yes

Reviewer #2: Yes

5. Review Comments to the Author

Reviewer #1: It was very interesting to read a clear description of the trends in Japanese cancer centers over the past 3 years after ID consultation and ASP intervention.

Full Title, "impact" is too strong for the data presented." I think "effects" is more appropriate.

Make sure bacterial genus and species names are italicized.

Please check the grammar of the manuscript for minor errors.

I think it is necessary to present the number of samples and isolates in each figure. Please consider adding this information.

line 69: Please add "CPE" abbreviation.

line 145: Enterobacterales should be changed to Enterobacteriaceae, because Enterobacterales and Enterobacteriaceae are synonymous.

line 325-328: I think it would be a good idea to mention mortality from infectious diseases.

line 367-370: CRE has a low isolation frequency to begin with, so it is possible that an increase in the number of specimens could increase the incidence. ESBL-producing Enterobacteriaceae is a common resistant organism, so the assessment that the detection rate increased due to underestimation before the intervention may not be true, but different factors may have been involved. Please add a discussion.

line 441: Reference has blank #5. Please delete it.

line 472: #14 is missing from the reference number.

Table 1: "F Fitting dose": "You should replace "and" with "or" in "function and other factors".

Reviewer #2: Authors demonstrate the possible casual impacts of the antimicrobial stewardship program and infectious disease consolation on the monthly days of therapy with antibiotics. Overall, the manuscript is well written and it provides helpful insight into understanding of epidemiological trends and the evaluation of the possible effects of the countermeasures. However, there are some aspects of the manuscript that could be made clearer, and I hope this suggestion make this manuscript to have the highest impact.

1. Days of therapy (DOT)

In the present study, authors defined the usage of antibiotics with the DOT per month and per 100 patient-days. However, I am wondering whether the amounts or types (e.g., oral or IV) of antibiotics were almost identical over time and over patients. Because if the variations in the used amount of antibiotics are not negligible, the calculation using the DOT may provide some biases, although it still can act as an indirect indicator. If so, I would like to suggest introducing the concept of the defined daily dose (DDD) to fully consider the amount of antibiotics in each patient.

2. Interpretation of changes in shape and slope in the ITS analyses

In the interrupted time-series (ITS) analyses, changes in the shape indicate the immediate effect of intervention, while the sustained effect can be represented by changes in the slope. Thus, although authors asserted that the estimated level is expected to decrease in the long-term observation (L340), the duration of observation will not affect the significance of changes in the shape. If authors tried to emphasize the expected sustained decreases in the DOT based on the significant decreases in the slope, I would like to suggest that authors can rephrase the sentence for the naïve readers.

Furthermore, according to the figures (especially in Figure 1 and Figure 3), the trends of DOT were started to be mainly changed from the late 2019, regardless of the start of intervention was April 2020. Thus, I would like to suggest that authors can clarify the possible reasons of this changes in the manuscript. In addition, if authors think the start point of visual impact of countermeasures can be delayed (e.g., delayed by a certain period of time-gap from the official start time of intervention), I would like to suggest considering a sensitivity analysis by varying the start of intervention time (adjusted start time considering the time-gap), since it may change the estimated changes in the level.

---

## [Author Response · Author response to Decision Letter 0]

17 Nov 2021

Our point-by-point responses to the comments and suggestions by Reviewer 1 are listed below.

We thank the reviewer for reviewing our manuscript and for the insightful comments that have helped us significantly improve our manuscript. Please note that our changes in the manuscript are highlighted in yellow.

Comment 1:

Full Title, "impact" is too strong for the data presented." I think "effects" is more appropriate.

Response:

We wish to thank the reviewer for this relevant comment. Accordingly, we have changed the title in the revised manuscript as follows:

“Effects of infectious disease consultation and antimicrobial stewardship program at a Japanese cancer center: an interrupted time-series analysis” (Title page, Page 1, Lines: 1-3).

Comment 2:

Make sure bacterial genus and species names are italicized.

Response:

We thank and agree with the reviewer’s pertinent comment. After careful review, we ensured that all bacterial genus and species have been italicized. (Page 2, Line: 41-42)

Comment 3:

Please check the grammar of the manuscript for minor errors.

Response:

We thank the reviewer for this valuable comment. We have carefully reviewed the manuscript to make sure there are no grammatical errors.

Comment 4:

I think it is necessary to present the number of samples and isolates in each figure. Please consider adding this information. 

Response:

We thank the reviewer for this important comment. Accordingly, we have added the following sentences:

“The number of isolated samples in 2018, 2019, and 2020 was 7, 9, and 13, respectively” (Results, Fig 4, Page 13, Line: 306).

“The number of isolated samples in 2018, 2019, and 2020 was 61, 72, and 69, respectively” (Results, Fig 5, Page 13, Line: 317).

“The number of isolated samples in 2018, 2019, and 2020 was 10, 5, and 8, respectively” (S1 Fig.).

“The number of isolated samples in 2018, 2019, and 2020 was 20, 24, and 39, respectively” (S2 Fig.).

“The number of isolated samples in 2018, 2019, and 2020 was 8, 12, and 20, respectively” (S3 Fig.).

“The number of isolated samples in 2018, 2019, and 2020 was 9, 8, and 3, respectively” (S4 Fig.).

Comment 5:

line 69: Please add "CPE" abbreviation.

Response:

We thank the reviewer for this relevant comment. We have changed the following text in the revised manuscript:

“Carbapenemase-producing Enterobacteriaceae (CPE) outbreaks have been reported in cancer centers across Japan, complicating infection control and treatment of infected patients [11]” (Introduction, Page 3, Line: 67-69).

Comment 6:

line 145: Enterobacterales should be changed to Enterobacteriaceae, because Enterobacterales and Enterobacteriaceae are synonymous.

Response:

We thank the reviewer for this insightful comment. We have changed the following texts in the revised manuscript:

“The resistant microorganisms included CPE, CRE, multidrug-resistant P. aeruginosa (MRPA), MRSA, and extended-spectrum beta-lactamase (ESBL)-producing Enterobacteriaceae” (Methods section, Page 8, Lines: 162-163). 

“S1 Fig. Trends of the incidence of carbapenem-resistant Enterobacteriaceae (CRE) per 1000 patients, by month, during Phase 2 of the intervention period” (Supporting information, Page: 24, Lines: 553-554). 

“S2 Fig. Trends of the incidence of extended-spectrum beta-lactamase (ESBL)-producing Enterobacteriaceae per 1000 patients, by month, during Phase 2 of the intervention period” (Supporting information, Page: 24, Lines: 555-556).

“Trends of the incidence of carbapenem-resistant Enterobacteriaceae (CRE) per 1000 patients, by month, during Phase 2 of the intervention period” (S1 Fig.).

“Trends of the incidence of extended-spectrum beta-lactamase (ESBL)-producing Enterobacteriaceae per 1000 patients, by month, during Phase 2 of the intervention period” (S2 Fig.).

Comment 7:

line 325-328: I think it would be a good idea to mention mortality from infectious diseases. 

Response:

We agree with the reviewer; however, we are unable to evaluate the infectious diseases mortality rate because we did not collect individual-level patient data in this study. Accordingly, we have added the following sentence:

“For more accurate assessment in future, it is thought that mortality due to ID should also be evaluated, but individual patient’s data were not evaluated in this study.” (Discussion section, Page 16, Lines: 378-379).

Comment 8:

line 367-370: CRE has a low isolation frequency to begin with, so it is possible that an increase in the number of specimens could increase the incidence. ESBL-producing Enterobacteriaceae is a common resistant organism, so the assessment that the detection rate increased due to underestimation before the intervention may not be true, but different factors may have been involved. Please add a discussion.

Response:

We thank the reviewer for the pertinent comments. The increase in the number of cefotaxime resistant E. coli and K. pneumoniae, which are suspected to be ESBL-producing, has been increasing yearly in Japan, and the increase in the total number of specimens was considered to be closer to the true nosocomial epidemiology. Regarding other factors, no nosocomial outbreaks due to CRE and ESBL-producing Enterobacteriaceae were observed during the study period. We have added the following texts in the revised manuscript:

“Especially in Japan, the incidence of cefotaxime resistant E. coli and K. pneumoniae, which are suspected to be ESBL-producing, has been increasing every year [2]. No in-hospital outbreaks of CRE or ESBL-producing Enterobacteriaceae were observed during the study period.” (Discussion section, Page 17, Lines: 400-402).

Comment 9:

line 441: Reference has blank #5. Please delete it.

Response:

We thank the reviewer for this astute comment. This was not a blank#5, it is the last page number for reference 3 on the reference list. It is as follows: “3. Brown AE. Overview of fungal infections in cancer patients. Semin Oncol. 1990;17 Supplement 6: 2-5” (Reference list, Page 21, Lines: 476-477).

Comment 10:

line 472: #14 is missing from the reference number.

Response:

We thank the reviewer for this important comment. We have added the missing reference number in the revised manuscript:

“14. WHO Collaborating Centre for Drug Statistics Methodology. ATC/DDD Index 2021 [Cited 2021 November 5]. Available from: http://www.whocc.no/atcddd/.” (Reference list, Page: 22, Lines 508-509).

Comment 11:

Table 1: "F Fitting dose": "You should replace "and" with "or" in "function and other factors".

Response:

We thank the reviewer for this relevant comment. Accordingly, we have changed the sentence as follows:

“The dose and method of administering the antimicrobials are inappropriate due to the renal function or other factors; thus, adjustments are necessary” (Page: 6, Table 1).

Our point-by-point responses to the comments and suggestions of Reviewer 2 are listed below.

We thank the reviewer for reviewing our manuscript and for the insightful comments that have helped us significantly improve our manuscript. Please note that our changes in the manuscript are highlighted in yellow.

Comment 1:

In the present study, authors defined the usage of antibiotics with the DOT per month and per 100 patient-days. However, I am wondering whether the amounts or types (e.g., oral or IV) of antibiotics were almost identical over time and over patients. Because if the variations in the used amount of antibiotics are not negligible, the calculation using the DOT may provide some biases, although it still can act as an indirect indicator. If so, I would like to suggest introducing the concept of the defined daily dose (DDD) to fully consider the amount of antibiotics in each patient.

Response:

We wish to thank the reviewer for these important comments. We calculated the AUD using DDD in addition to the DOT data, following the reviewer's suggestion, and performed additional analysis. The results of the analysis using AUD were the same as the results of the analysis using DOT. Accordingly, we have added sentences as follows:

“The results of these analyses of antimicrobial use are consistent with the DOT-based data based on antimicrobial use density (AUD).” (Abstract: Page 2, Lines 39-40).

“Antimicrobial use density (AUD) for carbapenem, three antipseudomonal agents, four narrow-spectrum antimicrobials

 AUD was also calculated to normalize the amount of each antimicrobial consumption. AUD was expressed as defined daily dose (DDD) per 100 patient-days for individual antimicrobial agent (AUD ＝antimicrobial consumption/DDD×100/patient-days). DDD per 100 inpatient-days for each drug or drug category prescribed monthly was calculated following the 2021 World Health Organization (WHO) Anatomical Therapeutic Chemical (ATC) classification system [14]. AUDs were calculated for carbapenem (CAR-AUD; for imipenem–cilastatin, meropenem, and doripenem); three antipseudomonal agents (piperacillin–tazobactam, cefepime, and cefozopran); and four narrow-spectrum antimicrobials (ampicillin, ampicillin–sulbactam, cefazolin, and cefmetazole).“ (Methods section, Pages 7-8, Lines: 147-156).

“To demonstrate the effect of each intervention on CAR-DOT, CAR-AUD, the DOT, and AUD for the three antipseudomonal agents (piperacillin–tazobactam, cefepime, and cefozopran) and four narrow-spectrum antibiotics (ampicillin, ampicillin–sulbactam, cefazolin, and cefmetazole) were ascertained. Furthermore, we carried out a segmented regression analysis of interrupted time-series studies for the two periods (Phase 1: April 1, 2018 to March 31, 2020; Phase 2: April 1, 2020 to March 31, 2021), when the DOT data were available.” (Statistical analysis, Page 9, Lines: 201-205).

“The trend in the CAR-AUD decreased along with its level (trend change, coefficient: -0.10; 95% CI: -0.15 to 0.05, p<0.001; change in level, coefficient: -0.60; 95% CI: -0.98 to -0.23, p<0.001).” (Results, Page 10, Lines: 233-234).

“The level of the monthly AUD for the three antipseudomonal agents did not decrease (coefficient: -0.01; 95% CI: -0.75 to 0.73, p=0.98), although its trend reduced (coefficient: -0.12; 95% CI: -0.22 to -0.031, p=0.01).” (Results, Page11, Lines: 250-252).

“The trend in the monthly AUD of the four narrow-spectrum antibiotics increased along with its level (trend change, coefficient: 0.43; 95% CI: 0.36 to 0.51, p<0.001; change in level, coefficient: 1.45; 95% CI: 0.85 to 2.04, p<0.001).” (Results, Page 12, Lines: 268-270).

“The results of the analysis between antimicrobial use using AUD and DOT were consistent.” (Discussion, Page 16, Lines: 374-375).

Comment 2:

In the interrupted time-series (ITS) analyses, changes in the shape indicate the immediate effect of intervention, while the sustained effect can be represented by changes in the slope. Thus, although authors asserted that the estimated level is expected to decrease in the long-term observation (L340), the duration of observation will not affect the significance of changes in the shape. If authors tried to emphasize the expected sustained decreases in the DOT based on the significant decreases in the slope, I would like to suggest that authors can rephrase the sentence for the naïve readers.

Response:

We wish to thank the reviewer for these insightful comments. Accordingly, we have added the following sentences:

“The levels of the three antipseudomonal agents did not decrease significantly, but the results suggest that a decrease can be expected in the future if significant decrease in trend is sustained” (Discussion, Page16, Lines: 366-368).

Comment 3:

Furthermore, according to the figures (especially in Figure 1 and Figure 3), the trends of DOT were started to be mainly changed from the late 2019, regardless of the start of intervention was April 2020. In addition, if authors think the start point of visual impact of countermeasures can be delayed (e.g., delayed by a certain period of time-gap from the official start time of intervention), I would like to suggest considering a sensitivity analysis by varying the start of intervention time (adjusted start time considering the time-gap), since it may change the estimated changes in the level.

Response:

We wish to thank the reviewer for these astute comments. The reason for the change was assumed to be due to the influence of an approximately one-hour lecture of an external infectious disease specialist on appropriate antimicrobial use for healthcare professionals on February 19, 2019. Sensitivity analysis was not performed because apparently, our active intervention was from April 2020. Accordingly, we have added the sentences as follows:

“Our interventions began in April 2020, but changes have been occurring in the use of carbapenems and the three anti-pseudomonal agents since late 2019. This was partly due to the impact of a one-hour lecture on the appropriate use of antimicrobials to healthcare providers by an external ID expert on February 19, 2019.” (Discussion, Page 17, Lines 405-407).

---

## [Decision Letter · Decision Letter 1]

12 Jan 2022

Effects of infectious disease consultation and antimicrobial stewardship program at a Japanese cancer center: an interrupted time-series analysis

PONE-D-21-27727R1

Dear Dr. Itoh,

We’re pleased to inform you that your manuscript has been judged scientifically suitable for publication and will be formally accepted for publication once it meets all outstanding technical requirements.

Kind regards,

Hiroshi Nishiura

Academic Editor

PLOS ONE

Additional Editor Comments (optional):

Reviewers' comments:

Reviewer's Responses to Questions

**Comments to the Author**

1. If the authors have adequately addressed your comments raised in a previous round of review and you feel that this manuscript is now acceptable for publication, you may indicate that here to bypass the “Comments to the Author” section, enter your conflict of interest statement in the “Confidential to Editor” section, and submit your "Accept" recommendation.

Reviewer #1: All comments have been addressed

Reviewer #2: All comments have been addressed

2. Is the manuscript technically sound, and do the data support the conclusions?

Reviewer #1: Yes

Reviewer #2: Yes

3. Has the statistical analysis been performed appropriately and rigorously? 

Reviewer #1: Yes

Reviewer #2: Yes

4. Have the authors made all data underlying the findings in their manuscript fully available?

Reviewer #1: Yes

Reviewer #2: Yes

5. Is the manuscript presented in an intelligible fashion and written in standard English?

Reviewer #1: Yes

Reviewer #2: Yes

6. Review Comments to the Author

Reviewer #1: (No Response)

Reviewer #2: All my comments were nicely addressed in the revised manuscript, and I was pleased to note that the additional analysis using DDD showed almost same results with those obtained using DOT.

7. PLOS authors have the option to publish the peer review history of their article (what does this mean?). If published, this will include your full peer review and any attached files.

Reviewer #1: No

Reviewer #2: No

---

## [Editor Report · Acceptance letter]

14 Jan 2022

PONE-D-21-27727R1 

Effects of infectious disease consultation and antimicrobial stewardship program at a Japanese cancer center: an interrupted time-series analysis 

Dear Dr. Itoh:

I'm pleased to inform you that your manuscript has been deemed suitable for publication in PLOS ONE. Congratulations! Your manuscript is now with our production department. 

Kind regards, 

on behalf of

Dr. Hiroshi Nishiura 

Academic Editor

PLOS ONE